# Influence of Polyurea Coatings on Low-Longitudinal-Reinforcement-Ratio Reinforced Concrete Beams Subjected to Bending

**DOI:** 10.3390/ma15072652

**Published:** 2022-04-04

**Authors:** Jacek Szafran, Artur Matusiak, Katarzyna Rzeszut, Iwona Jankowiak

**Affiliations:** 1Department of Structural Mechanics, Faculty of Civil Engineering, Architecture and Environmental Engineering, Lodz University of Technology, Aleja Politechniki 6, 90-924 Łódź, Poland; artur.matusiak@dokt.p.lodz.pl; 2Institute of Building Engineering, Faculty of Civil and Transport Engineering, Poznan University of Technology, Piotrowo 5, 60-965 Poznań, Poland; katarzyna.rzeszut@put.poznan.pl; 3Institute of Civil Engineering, Faculty of Civil and Transport Engineering, Poznan University of Technology, Piotrowo 5, 60-965 Poznań, Poland; iwona.jankowiak@put.poznan.pl

**Keywords:** reinforced concrete beams, bending elements, serviceability limit state, polyurea, cracks, durability of reinforced concrete elements, reinforcement, retrofitting

## Abstract

“Polyurea coatings as a possible structural reinforcement system” is a research investigation that aims to explore the possible applications of polyurea coatings for improving structural performance (including steel, concrete, timber and other structures used in the construction industry). As part of the research in this field, this paper focuses on evaluating the performance of bending polyurea-coated reinforced concrete (RC) beams with a low reinforcement ratio. The easy application and numerous advantages of polyurea can prove very useful when existing RC structural elements are repaired or retrofitted. Laboratory tests of RC beams were performed for the purpose of this paper. The failure mechanisms and cracking patterns of these specimens are described, and their bending strengths were compared. On this basis, the effect of the coating on bending strength and the performance of the reinforced beams at the serviceability limit state (SLS) was examined and analyzed. The results showed that the use of a polyurea coating has a positive impact on the cracking and deflection state of RC beams and makes it possible to safely use RC elements on a continuous basis under high levels of load.

## 1. Introduction

Concrete and reinforced concrete (RC) elements are commonly used today by building engineers to construct both traditional buildings and bridge structures, including various and even very unusual structural arrangements. Since these elements are frequently chosen and sometimes used under severe operating conditions, research work has broadened, and new materials have been introduced to produce and protect such parts of building facilities.

RC structures that are correctly designed and produced tend to be relatively durable. It is commonly assumed that the core structural elements that are critical for the safety of a building facility should serve their purposes throughout a building’s whole service life. Therefore, the durability of RC beams and pre-cast concrete products is a prerequisite for the durability of a whole building. The increasing loads exerted on RC elements, crack formation and propagation, and the adverse effects of aggressive media may all result in the ultimate limit state (ULS) and the serviceability limit state (SLS) being exceeded. These phenomena are indicated in RC structures by excess cracking and the deflections of such elements. The cracking state of RC elements is particularly important as it is the main factor determining the durability and safety of structures. As a matter of fact, crack formation is inherent in the performance of RC structures as a natural response of the material to the state of strain to which it is subjected. Nevertheless, the development of wide cracks is very disadvantageous, as they provide good conditions for penetration by fluids that cause the corrosion of rebars and affect the rate of concrete degradation, decreasing the durability of RC elements. With respect to cracking, the safety of structures can be improved by certain design measures (when new elements are designed) or by repairing existing RC elements by means of filling and closing the cracks. The process of repairing RC structures is often labor-intensive, expensive and difficult to complete during the operation of a building facility [1,2].

RC beams are designed with specific assumptions concerning their operation such as certain ambient conditions and external loads. This is why excess changes in load and operating conditions can produce local failures or even jeopardize the safety of the whole structure. Today’s engineering know-how makes it possible to find solutions to improve the performance of existing RC elements. Such modifications are made by introducing additional external reinforcing elements (such as steel sections or carbon fiber tapes) and fixing the cracks. Undoubtedly, one disadvantage of these solutions is that they both require much work and sufficient space around an RC element to allow for the assembly of reinforcements [3,4].

An alternative solution consists of applying a polyurea coating on the external surfaces of RC beams, which certainly requires less space around the reinforced elements when compared to the above-mentioned solutions. Although this system mainly protects concrete against external factors, it can also improve the service life, elasticity, and bending strength of concrete elements [5,6,7]. The main advantage of polyurea application is the fast process of preparing the substrate and applying the membrane that effectively protects an RC element. This technological solution can significantly reduce the time needed to repair a structure and the effort required to improve the performance of existing RC structures [5,6,7].

A polyurea coating system was invented in the 1980s in the United States and was soon used in Europe. Polyurea is the reaction product of an isocyanate component and a resin blend component. The final product has a chain structure forming a durable and elastic material with various possible applications in the construction industry. Polyurea coatings have been commonly used to protect concrete and RC structures against the adverse impacts of water and corrosion and to make them leak-tight [5,6,7]. Experimental research on this material, which started at the turn of the 21st century, focused on analyzing basic properties of polyurea. Papers [8,9,10,11,12] address the basic properties of the coating, mainly its elasticity in tensile tests. Papers [13,14,15,16] analyze polyurea properties in variable ambient conditions (such as high temperature). In another group of papers, possible applications of polyurea and composite materials in ballistic systems and equipment are described. Papers [17,18,19,20,21] focus on analyzing the effect of a polyurea coating on components of ballistic equipment (such as protective helmets or aluminum and steel plates). Papers [22,23,24,25,26] describe how composite (polymer) materials protect specific structural components against the effects of an explosion. In contrast to numerous studies on the properties of polyurea coatings, only a small number of papers can be found in the literature that focus on the use of polyurea to improve the performance of structural elements. Papers [27,28,29,30,31,32,33] summarize the results of studies on the effect of polyurea applications in the performance properties of certain components used in the construction industry (such as wooden connections, water pipes, steel plates, and concrete rings). However, the available papers provide no general explanation as to how the use of a polyurea coating changes the performance properties of structural elements. The present paper aims to provide more knowledge on this matter.

The proposed method of covering a reinforced concrete beam with a polyurea-layered coating does not significantly increase the bending capacity, but it enables an improvement in the performance characteristics of a reinforced concrete beam in its normal and exceptional conditions, which ensures the possibility of the cyclic loading of beam elements without visible loss of load capacity.

This paper proposes a new way of improving the performance properties of bent RC beams by applying a polyurea coating. Tests were carried out to analyze how the use of an external layer of a polyurea coating influences the bending strength of RC beams. The effect of a polyurea coating on the failure mechanism of each specimen was also examined. It was stated that the use of a polyurea coating on the surfaces of reinforced concrete beams allowed for the safe relief of the structure after a load exceeding its limit load capacity. Such a situation may take place when the structure is operating in a failure state. Thus, the use of a polyurea coating can significantly improve the safety of the entire structure and its users. The new characteristics of the bending element, obtained in this way, enable the element to be used after exceeding the permissible deformations, while maintaining an acceptable level of safety.

## 2. Materials and Methods

### 2.1. Materials

#### 2.1.1. Polyurea

Polyurea is the reaction product of two components mixing at a high temperature (between 65 °C and 80 °C) and at a high pressure (between 120 bar and 200 bar) in certain proportions. Polyurea is an elastomer that is derived from the chemical reaction (polyaddition) of an aromatic or aliphatic isocyanate component and a multifunctional amine or an amine blend. Aromatic polyureas are derived from methylenediphenyl diisocyanate (MDI), while aliphatic polyureas are derived from hexamethylene diisocyanate (HDI) or isophorone diisocyanate (IPDI), which form a stiff chain section. The precision of mixing and dosing the components plays a prominent role in the spray showering process of polyurea. The parameters of spraying (temperature and pressure in the device) have to be rigorously controlled; they have to comply with the norms given by the manufacturer in the product data sheet (DPS). Moreover, the appropriate voluminal and weight ratios of the components have to be continuously controlled before and during the process of spraying. The final product has a very short time of bonding and high levels of chemical and water resistance, and elasticity. Polyurea exhibits a very high adhesion to many materials (such as steel, plastics, wood, and concrete) [5,6,7].

The tests were performed using aromatic polyurea as the most common type of coating utilized in the construction industry. The components supplied were used to produce specimens in order to determine strength properties of polyurea and to apply them on RC beams. The mechanical properties of the coating were obtained in a static tension test according to EN ISO 527:2012 [34]. The tension tests on dumbbell-shaped specimens were performed with two test speeds of 50 mm/min and 100 mm/min. All the tension tests were conducted using the INSTRON 5582 tensile tester (INSTRON, Norwood, USA). Lengths of a polyurea specimen at the beginning of the tension test and before breaking can be seen in Figure 1. The results of the polyurea tension tests are listed in Table 1. The coating tensile strength was 24.08 MPa with an engineering strain of 417% at a test speed of 50 mm/min and 23.03 MPa with an engineering strain of 391% at the test speed of 100 mm/min (Table 1). The analysis of the results showed that the tensile strength and the nominal engineering strain of the membrane depend on the test speed, and these properties are lower at higher specimen loading speeds.

#### 2.1.2. Concrete

The concrete mix used in the tested beams was made of pit sand, gravel of grain size 2–8 mm and 8–16 mm, Portland cement CEM I 42.5 (pure Portland cement without any additives – class of 42.5), water, and certain concrete admixtures. The main components of the mix per 1 m^3^ of concrete are listed in Table 2.

When the RC beams were concreted, six cube-shaped specimens of side 150 mm were produced. The specimens were used to determine strength properties of the concrete. Compression strength of the concrete was obtained according to EN 12390-3:2019 [35], and its tensile splitting strength according to EN 12390-6:2011 [36]. The results of the concrete strength tests are listed in Table 3.

#### 2.1.3. Reinforcing Steel

The upper longitudinal reinforcement of the RC beams was made of #10 mm steel rebars (B 500 B), the lower one made from #14 mm steel rebars (B 500 B), and the transverse reinforcement in the form clevises made from #6 mm rebars (B 500 B). All rebars were made of A-III ribbed steel. Specimens of each type of rebars were taken to determine their strength properties according to EN ISO 15630-1:2019 [37]. The results of mechanical tests of reinforcing steel are listed in Table 4.

### 2.2. Reinforced Concrete Beams

The six RC beams made of concrete characterized in Section 2.1.2 were subjected to laboratory bending tests. The RC beams were reinforced with two #10 mm rebars in the compression area (the upper one) and with two #14 mm rebars in the tension area (the lower one). The transverse reinforcement was made of #6 mm rebars in the form of clevises with the main spacing of 15 cm at the beam midspan, and with spacing of 10 cm in the support area. The dimensions and arrangement of the reinforcement used in the RC beams are shown in Figure 2 and Figure 3.

The bending reinforcement ratio (*ρ_s_*) is defined as the ratio of the area of the longitudinal reinforcement under tension (*A_s_*) to the cross-sectional area of the bent beam (*A_c_*) by formula:*ρ_s_* = *A_s_*/*A_c_*(1)

According to this definition, the bending reinforcement ratio of the RC beams is *ρ_s_* = 0.7%, which is why they can be classified as elements with a low longitudinal reinforcement ratio.

All RC beams were divided into two batches. Three specimens comprising the first batch were marked as control specimens and had no polyurea coating. Three specimens of the second batch were polyurea-coated on all of their outer surfaces. The characteristics of specimens used in the tests can be found in Table 5.

### 2.3. Beams with Polyurea Outer Layer

The three specimens of the second batch were polyurea-coated on all of their outer surfaces. The diagram and the coated elements are shown in Figure 4.

The polyurea application process involved three main phases: surface preparation of the RC beams, prime coat application, and polyurea coating application. The process of preparing the RC elements for tests is shown in Figure 5.

The phase when the beams’ surfaces were prepared before applying polyurea involved checking the elements in terms of the substrate stability, surface defects, and the presence of substances inhibiting the coating adherence. After the specimens were checked, the concrete surface was polished to remove cement wash and any loose particles (Figure 5b).

During the second phase, a specific epoxy-resin-based primer was applied, and dry quartz sand was sprinkled, to achieve the best coating adhesion (Figure 5c). Before and during prime coat application, substrate and ambient temperatures as well as humidity were controlled to provide optimal conditions for this process.

Spray coating was applied in two layers: the first layer directly on the concrete surface and the second one directly on the first layer, perpendicular to the direction in which the first one layer applied (Figure 5d). This method of application made it possible to provide a fully leak-tight, continuous layer with an average thickness of 2.5–3.0 mm, according to the material manufacturer’s guidelines. During the polyurea spraying process, the temperature of ingredients was checked by special technical equipment (aggregate) and its optimum was +75 °C. Moreover, the temperature of concrete’s surface and air was verified by handheld thermometer, and its optimum temperature was +15 °C.

### 2.4. Test Stand

Both batches of the RC beams were tested on one test stand presented in Figure 6 and Figure 7. The test stand consisted of the following components:A steel main frame of the test stand;A steel frame supporting the RC beams;A stand-alone steel frame supporting strain sensors;A hydraulic piston mounted to the upper part of the frame;A hydraulic pump to drive the piston;A workstation for data acquisition.

The RC beams were loaded by a steel traverse, which was symmetrically oriented to the beam’s perpendicular axis and produced load in the form of two concentrated forces 100 cm apart. Force was applied to the steel traverse via a centrally located hydraulic piston. The hydraulic piston applied the appropriate pressure by means of a hydraulic pump. The pressure was increased in steps to obtain the specified value of the force in the piston. Reinforced concrete beams were loaded with step force increments of 5 kN or 10 kN. A force gauge was mounted between the hydraulic piston and the steel traverse in order to double verify the current value of the load on the tested elements. The test specimens were positioned on the horizontal steel frame and pivotally supported at two points 320 cm apart along the axis.

The test stand was also equipped with 12 sensors to measure the displacement of the RC beams while they were subjected to load. The set of 12 sensors included 10 inductive sensors and 2 dial gauges (marked as S-1 and S-2). A force gauge was also mounted under one of the pivot supports to record responding changes under the beam. Sensor locations on the beam are shown in Figure 7.

### 2.5. Measurements

Vertical displacements, the force exerted by the hydraulic piston, and the response value at one of the supports were measured during the tests. Beam deflection was measured by means of inductive sensors (linear displacement transducers) fixed to the stand-alone steel frame (Figure 6). The readings from all the types of sensors were automatically recorded at each load level every 0.2 s. A workstation with necessary software was used to control the measurement process and record readings from each sensor. For selected beams, the cracking pattern on their surface was also captured by marking, near each crack, the load value at which it formed.

The beams were loaded by forces that increased in steps up to failure, which was the case with the uncoated beams, or by applying an unloading/loading cycle for the polyurea-coated beams. The value of the force was obtained from the device supplying the hydraulic piston and also checked by means of the force gauge mounted between the piston and the traverse.

## 3. Results

### 3.1. Bending Strength of RC Beams

The relation between the force (exerted by the hydraulic piston) and the midspan beam deflection is presented in Figure 8. In the initial phase of loading the samples in the range of 0–10 kN (Figure 8), a linear relationship between the force in the main hydraulic piston and deflection in the middle of the span can be noticed. This dependence is not directly proportional, which indicates the high toughness of reinforced concrete beams before the appearance of cracks in the concrete. Table 6 lists breaking forces, their average values, and gains compared with breaking forces obtained for the uncoated reference beams.

The average breaking force (bending strength) of the RC beams was calculated as the arithmetic mean of the three tests of each type of beams. Table 6 also includes the average gain in the load-carrying capacity of the polyurea-coated specimens over the reference ones. The load-carrying capacity gain was defined in (kN) and (%) as a difference between the destructive forces for the coated specimens and for the reference specimens.

The summary of breaking forces indicates that the breaking force is higher for the polyurea-coated RC beams than for the reference beams. Due to the application of the coating on the outer surface of the beams, the breaking force (bending strength) was higher by 4.7% (4.6 kN). The increase in the load-carrying capacity of the RC beams is objectively insignificant in terms of the material and labor costs related to membrane application.

The application of the polyurea coating on the RC beams made it possible to perform an unloading/loading cycle on the test specimens; the unloading point was set at 90% of the breaking force found for the uncoated reference beams. This is the crucial difference between the coated beams and the uncoated reference beams under load. When the load exerted on the specimens (P.2.1–P.2.3) reached 90% of the breaking force for the reference beams, it was relieved and restored; after that, the specimens exhibited the original bending strength (the strength before the unloading/loading cycle). The secondary strength (the bending strength after the unloading/loading cycle) of the polyurea-coated RC beams was achieved without any excess increase in the deflection of these elements, i.e., without any loss of bending stiffness (Figure 8).

### 3.2. Displacements of the RC Beams

The displacements of the RC beams, defined as the level of deflection of a specimen at measurement points, were measured on a continuous basis throughout each test.

The increase in load resulted in larger deflections but no side deformations and no torsion of the beams were observed during the tests (Figure 9). In line with a common behavior of RC beams, the specimens were deflected in the direction of the load applied (Figure 10). The relation between the midspan beam deflection and the load (the force exerted by the hydraulic piston) is shown in Figure 8.

In order to compare strains of the polyurea-coated RC beams with those of the reference ones, beam displacements along their longitudinal axes are shown in Figure 10. Displacements observed along the beams under the loads corresponding to 20% and 80% of the breaking force for the reference beams can be seen in Figure 10b,c, respectively.

The diagram (Figure 8) and Table 6 show that the maximum breaking forces for the coated specimens were obtained with similar deflections of the reference beams. This means that, although the unloading/loading cycles were performed, the bending stiffness of the specimens (P.2.1–P.2.3) was not lost, and their total displacements were close to strains of the reference specimens.

The analysis of the curves displayed in Figure 10 indicates that the values and curves of the displacements in five out of six of the RC beams tested are comparable. It can be seen that in the initial phase of loading, when beam stiffness mainly depends on the properties of concrete, the displacements of the coated beams are smaller than those of the reference specimens (Figure 10b,c). The differences in the curves of strains in the specimens disappear under a higher load, i.e., when the beam stiffness is related to characteristics of the reinforcing steel. The diagrams show that the displacements of beam B.2.1 are the largest (Figure 8 and Figure 10). This is due to the fact that this beam was the first one to be tested under a load after the test stand was built, so some strains of this beam are related to the settlement of the test stand structure. It should be noted that the results of the other specimens are stable and coherent.

### 3.3. Component Cracking

The cracking pattern was observed during the tests on a continuous basis using two cameras from both sides of the RC beams. For selected specimens, the cracking pattern on the surface of the elements was also captured by marking, near each crack, the load value at which it formed (Figure 11). For all the RC beams analyzed, typical bending cracks were recorded, and these tended to develop at the beam midspan.

Numerous vertical cracks were observed on the uncoated beams in the middle of the span (specimens B.2.1 to B.2.3). Such a system of cracks was forced by the pure bending zone, between two concentrated forces in accordance with the assumed static scheme of the experiment. The presence of multiple cracks on the surface of RC beams significantly increases, eventually reaching the serviceability limit state of these elements. The cracking on the surface of the uncoated RC beams is shown in Figure 12.

In the case of the polyurea-coated RC beams (specimens P.2.1 to P.2.3), the largest cracks were observed at the beam midspan. The polyurea coating efficiently covered the surface cracking to the extent that only wide cracks could be seen. The crack bridging by the coating generated a difference in the visual perception of the crack pattern on elements covered with polyurea coating in relation to uncoated beams (Figure 12 and Figure 13) because, in this case, smaller scratches were not noticeable to the naked eye. During the tests, it was found that the polyurea coating covered cracks of up to 5 mm in width in the cracked cross-sections. In this case, the concrete cracking state should not accelerate, eventually reaching the serviceability limit state as there are no open cracks on the surface of the beams. The polyurea coating efficiently covers cracks and protects the element against penetration by corrosive fluids (water, air, and chemical compounds). The cracking on the surface of the coated RC beam is shown in Figure 13.

### 3.4. Failure Mechanisms

All beams failed in a process typical for bent elements: the concrete was crushed (debonded) in the compression area of the beam cross-section, or the reinforcement yielded in the tension area of the beam cross-section (Figure 14). The reference beams (specimens B.2.1, B.2.2, and B.2.3) failed when the concrete debonded in the upper (compression) area of the elements’ cross-section. In the initial phases of imposing the load on the reference beams, vertical cracks appeared in the middle part of these beams; as the load increased, the cracks elongated and widened. Finally, in line with the characteristic failure mechanism of bent elements, the beams suddenly failed after exceeding the concrete strength in the upper part of the beam cross-section (Figure 14a).

In contrast to the reference beams, the polyurea-coated beams (specimens P.2.1, P.2.2, and P.2.3) failed when the reinforcement yielded in the lower (tension) area of these elements. In line with the characteristic failure mechanism of bent elements, after the bending strength of the RC beams was exceeded, they indicated failure and failed following the reinforcement yielding in the lower part of the cross-section (Figure 14b). This was accompanied by a rapid widening of vertical cracks near the point where the reinforcement yielded with the increasing beam deflections. The polyurea coating remained well-bonded to concrete and covered all cracks until the complete failure of the beams as a result of bending. The membrane efficiently constrained the concrete in the upper part of the beam cross-section, which is why a safer failure mechanism occurred in the form of reinforcement yielding in the tension area. The failure of a bent element as a result of the reinforcement yielding is believed to be safer as it is indicated by the excess strains and deflections of an RC beam.

### 3.5. Serviceability Limit State (SLS)

As defined in EN 1990:2004 [38], the serviceability limit state corresponds to certain conditions, which if they are exceeded, cause a structure or its element to fail to meet performance requirements. According to EN 1992-1-1:2008 [39], the serviceability limit states for RC structures include:Stress limitation;Crack control;Deflection control.

The verification of cracks and deflections in RC structures is of particular importance as these conditions should not impair the proper functioning and durability of the structure or cause its appearance to become unacceptable. The cracking and deflection of RC structures as a result of loading is normal in reinforced concrete structures subject to bending. Cracks may also arise from causes other than the load exerted on a structure, such as plastic shrinkage or expansive chemical reactions within the hardened concrete.

The application of a polyurea coating on the RC beams subject to bending had a positive impact on limiting the deflections of these elements in the initial loading phase, i.e., in conditions corresponding to normal operation. The cracking of the polyurea-coated elements (specimens P.2.1 to P.2.3) was significantly limited as cracks were efficiently covered by a highly elastic membrane. The analysis of the impact of a polyurea coating on beam cracking and strains may indicate that the polyurea coating can be used to improve the performance of bent RC beams with a low longitudinal reinforcement ratio at the serviceability limit state.

### 3.6. Description of Crack Formation in Polyurea-Coated Beams Based on Calculations

For bent RC elements, crack formation is usually described using three properties: cracking moment, crack width, and crack spacing. The cracking moment (*M_cr_*) causes crack formation and is defined as the bending moment at which the concrete tensile strength or, to be more specific, the level of critical strain is exceeded. This is often a random value since the concrete structure is not always homogenous and RC elements can be of low quality in terms of workmanship. The approximate interval of this value is difficult to clearly indicate, because it is a multi-parameter phenomenon. According to the theory concerning the phases of operation of a bent element, the cracking moment occurs between phases I and II of the member operation [40,41].

In [41], various ways of describing the cracking moment are analyzed. In order to provide calculations describing crack formation in polyurea-coated beams, two approaches to calculate the cracking moment were chosen:Without taking the longitudinal reinforcement of beams into account, using formula:
*M_cr1_* = *f_ctm_* × *W_c_*(2)Taking the longitudinal reinforcement and the equivalent section into account, using formula:
*M_cr2_* = *f_ctm_* × *W_cs_*(3)

Theoretical stress distribution in a bent RC beam cross-section at the transition point between phases I and II, i.e., when the first crack formed, is shown in Figure 15.

The component (*f_ctm_*) used in Equations (2) and (3) is the average concrete tensile strength; the section modulus of a rectangular cross-section subject to bending (*W_c_*) was calculated from the expression *b·h^2^/6*. The variable (*W_cs_*) is the section modulus of the equivalent section, taking into account the impact of reinforcement on the cross-sectional stiffness. To analyze the values of the cracking moment, the height of the tension area (*x*) immediately before crack formation in the beams must be determined. To this end, we need to calculate the position of the neutral axis in the equivalent section. First, the steel reinforcement has to be replaced with an equivalent area *α_e_A_s_*, where *α_e_* is the ratio of the modulus of elasticity of steel (*E_s_*) to the modulus of elasticity of concrete (*E_cm_*), expressed as:*α_e_* = *E_s_*/*E_cm_*(4)

Next, we calculate the equivalent section area (*A_cs_*), the static moment with respect to the upper edge of the cross-section (*S_cs_*), and finally, the height of the compression area is given by the formula:*x* = *S_cs_*/*A_cs_*(5)

Table 7 lists the results of calculations required to determine the height of the compression area.

In order to determine the section modulus (*W_cs_*) using Equation (6), we have to calculate the equivalent moment of inertia (*I_cs_*) with respect to the axis passing through the center of gravity of the equivalent section (*A_cs_*):*W_cs_* = *I_cs_*/(*h* − *x*)(6)

To analyze the cracking moments (*M_cr_*) for the uncoated beams, the average tensile splitting strength obtained during the in-house testing of concrete specimens (Table 3) was taken as the average tensile strength of concrete (*f_ctmC_*). For the polyurea-coated beams it is justified to consider the average tensile strength of the polyurea coating at the test speed of 100 mm/min (Table 1) to be the average tensile strength of concrete (*f_ctmP_*). All obtained results are listed in Table 8, and the actual stress diagram across the beam section is shown in Figure 16.

The obtained values of cracking moments for the polyurea-coated beams are over six times higher than those for the uncoated beams. The main reason for this is that polyurea has a much higher tensile strength than concrete. During the tests, it was observed that the polyurea coating did not break until the complete failure of the RC beams. This fact and the calculation analysis indicate that the polyurea coating has measurable benefits in terms of crack control at the serviceability limit state of bent RC beams.

### 3.7. Cost Analysis of Polyurea Coating Application on Reinforced Concrete Elements

The application of polyurea coatings on concrete surfaces is a complex process comprising three main phases:Preparation of an existing concrete surface;Appropriate priming of the concrete surface;Applying the polyurea coating.

The final cost of applying the polyurea coating largely depends on the humidity and quality of, and the level of damage to, the existing concrete surface. The cost of such a service is proportional to the scope of preparation processes required to achieve a proper condition of the surface so that an appropriate epoxy-resin-based primer can be applied. Estimated costs of polyurea coating applications, including surface preparation, are:From 75 EUR/m^2^ net for dry concrete surface without any defects and cracks;To 160 EUR/m^2^ net for wet concrete with deep defects and cracks.

Reinforced concrete elements are a specific type of structural component very often designed for specific conditions and the interior of buildings. This is why any repair of such components requires a lot of preparatory work and involves severe difficulties in the interior of buildings. The use of polyurea coatings to repair existing reinforced concrete elements and/or to improve their properties can prove a very cost-effective renovation method. This solution does not require that many works, and the components can be repaired inside buildings, which can be highly cost-effective both in financial and organizational terms.

## 4. Conclusions

The aim of this paper was to examine the impact of polyurea coatings on low-longitudinal-reinforcement-ratio reinforced concrete beams subjected to bending. Based on the conducted research, it was found that the application of polyurea coatings allowed for an increase in crack-bridging efficiency and enabled the improvement in the performance of a reinforced concrete beam in both normal and failure states. The application of a polyurea coating in a few layers made possible to relieve the beam elements without a visible loss of their load capacity and with a satisfactory level of crack bridging in case of periodical load application. In contrast to traditional isolating materials, polyurea membranes have excellent functional properties, chemical resistance, and mechanical strength. The material properties of polyurea that make it usable in a wide range of applications and are, therefore, its advantages, include: fast reactivity and bonding (time saving), adherence to most building materials, high mechanical strength, and effective crack bridging; experimental studies have shown that a coating can bridge cracks up to 5 mm in width. Therefore, the effect of this coating application on failure mechanisms, deflection and cracking patterns in the RC elements was of special interest. Based on the results of the experimental research and their analyses, the following conclusions were drawn:The application of polyurea coatings on the outer surface of RC beams increases their bending strength by 4.7%. The increase in the load-carrying capacity of the RC beams is insignificant in terms of the material and labor costs related to the membrane application;The tests showed that polyurea-coated RC beams can be safely subjected to loading and unloading (up to a value of 90%, which was used in the experimental research), which is the main difference between them and typical RC beams that cannot be safely subjected to this process;The impact of polyurea application on deflections of RC beams is visible at small load values, under normal operating conditions when the stiffness of an element mainly depends on the properties of concrete;The polyurea coating efficiently covers cracks and may protect the RC element during its exploitation against penetration by corrosive fluids (water, air, and chemical compounds); it brings considerable benefits because the coating may prolong the service life of the elements in such a manner;The tests revealed that elastic coating RC beams with a polyurea coating bonded well to concrete, providing full integrity of the elements throughout the whole loading process up to their complete failure;When the polyurea coating is applied, it creates considerable benefits when RC beams are used at the serviceability limit state because the coating covers cracks and prolongs the service life of the elements;The results of the experimental research and analytical calculations indicate that the polyurea coating can be used as a material designed to protect RC beams from cracking and the risk of exceeding their serviceability limit state due to cracking;The polyurea coating may improve the safety of people and RC structures as it replaces failure mechanisms with safer mechanisms and makes these elements integral and durable in imminent failure conditions.

## Figures and Tables

**Figure 1 materials-15-02652-f001:**
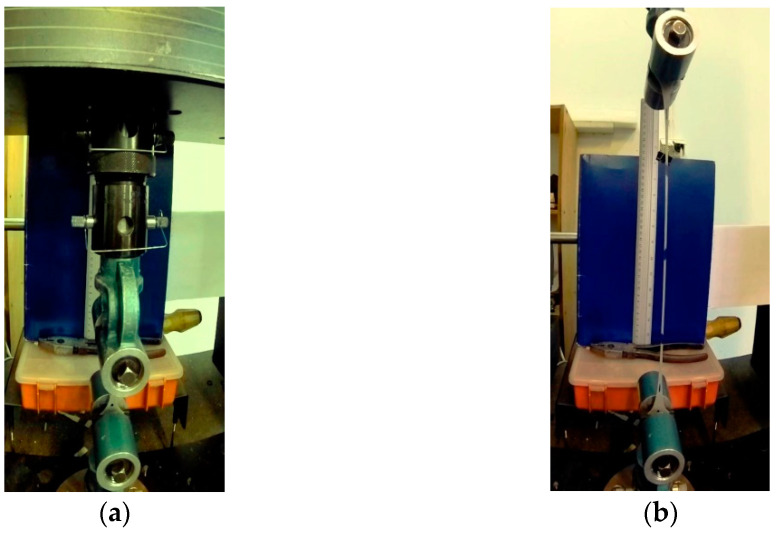
A polyurea specimen in the tensile tester: (**a**) At the beginning of the tension test; (**b**) before breaking.

**Figure 2 materials-15-02652-f002:**
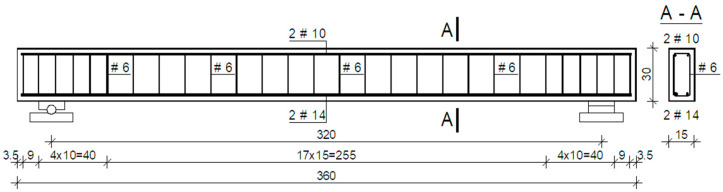
Dimensions and arrangement of rebars in the RC beams (dimensions in cm).

**Figure 3 materials-15-02652-f003:**
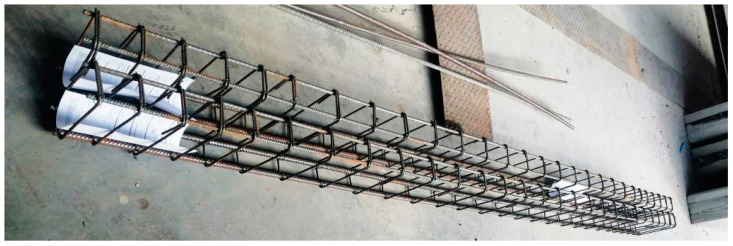
The arrangement of rebars for the RC beams before concreting.

**Figure 4 materials-15-02652-f004:**
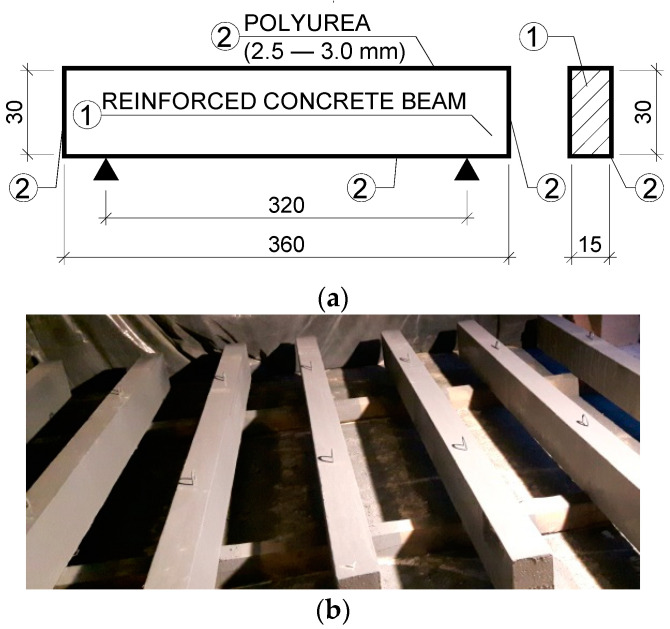
Polyurea-coated elements: (**a**) Coating arrangement (dimensions in cm); (**b**) polyurea-coated elements.

**Figure 5 materials-15-02652-f005:**
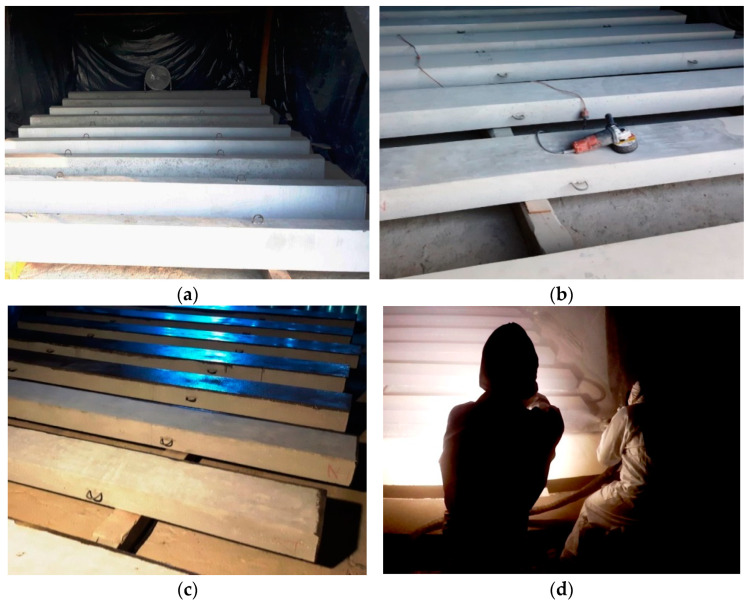
Polyurea application process: (**a**) Positioning the RC beams; (**b**) polishing the beams; (**c**) prime coat application; (**d**) polyurea application.

**Figure 6 materials-15-02652-f006:**
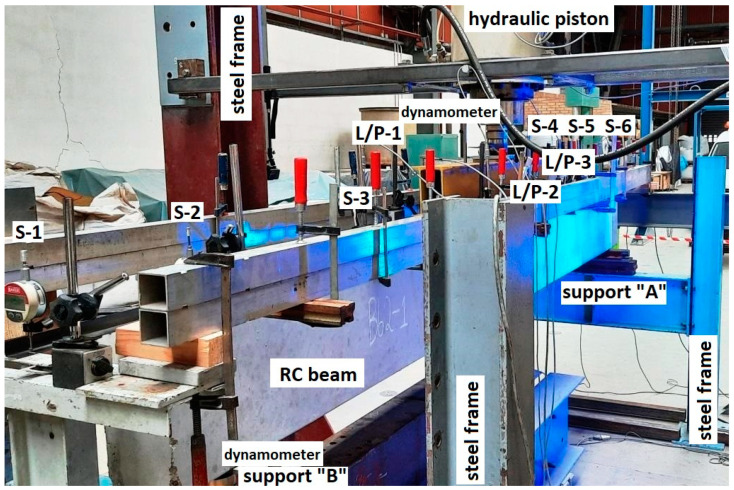
General view of the test stand.

**Figure 7 materials-15-02652-f007:**
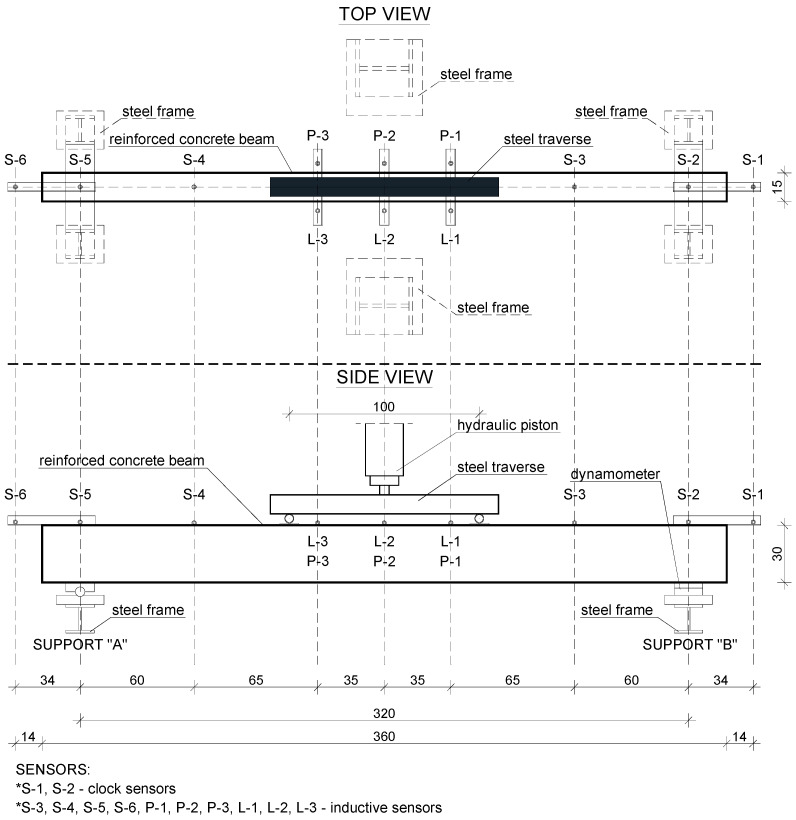
The test stand for testing RC beams.

**Figure 8 materials-15-02652-f008:**
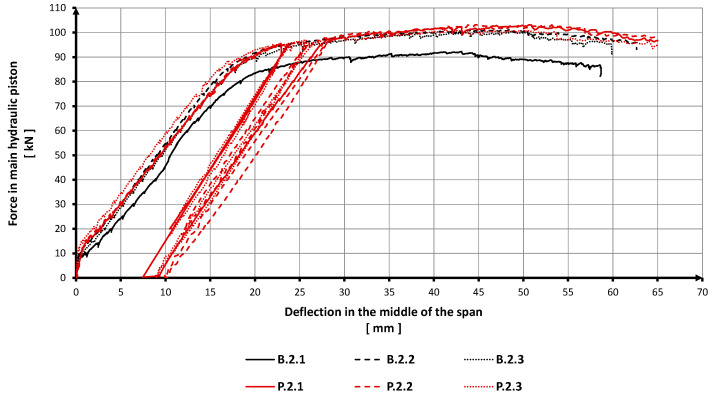
Force (exerted by the hydraulic piston) vs. beam deflection.

**Figure 9 materials-15-02652-f009:**
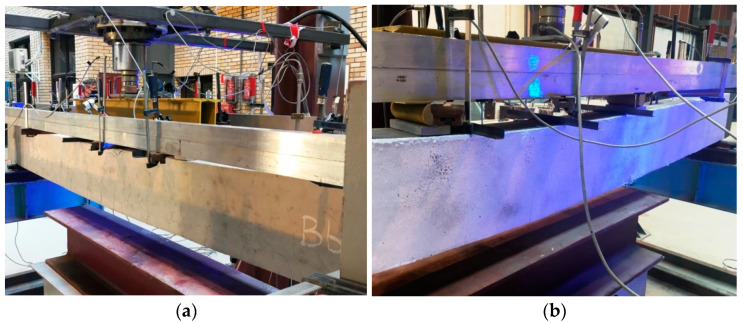
Beams under loading—views along the elements: (**a**) A beam without the polyurea coating; (**b**) a polyurea-coated beam.

**Figure 10 materials-15-02652-f010:**
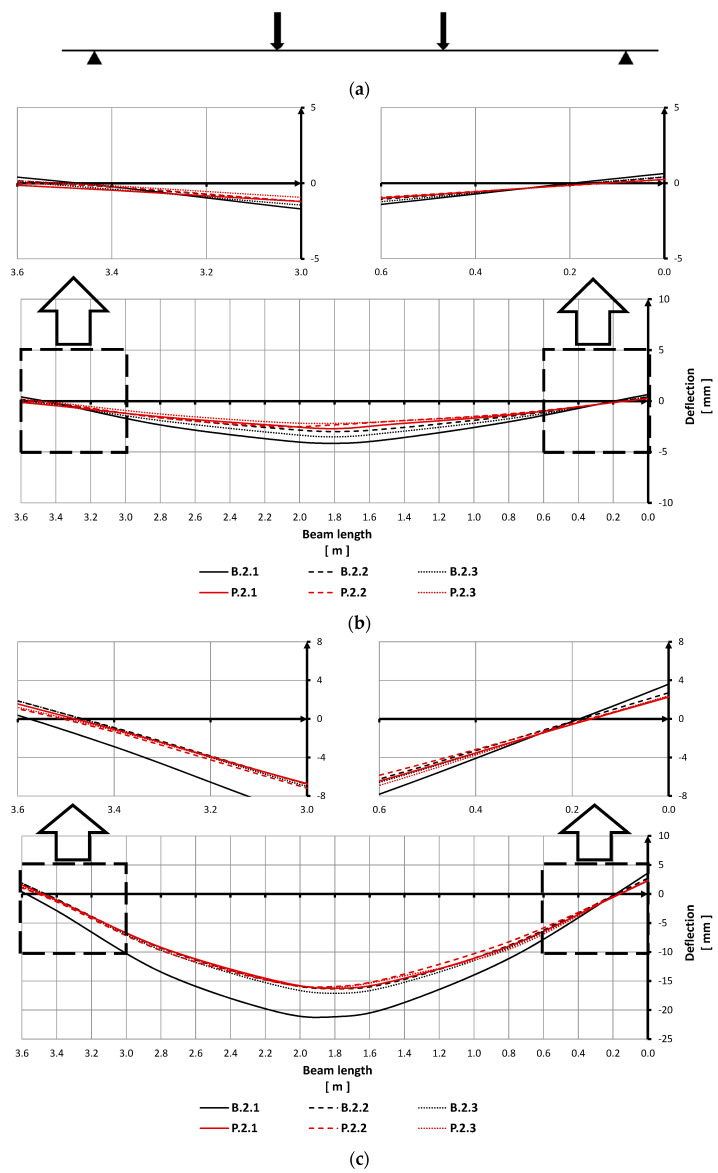
Beam displacements along the specimen under loads equal to 20% and 80% of the breaking force: (**a**) Arrangement of load exerted on the beam; (**b**) beam displacement under load equal to 20% of the breaking force; (**c**) beam displacement under load equal to 80% of the breaking force.

**Figure 11 materials-15-02652-f011:**
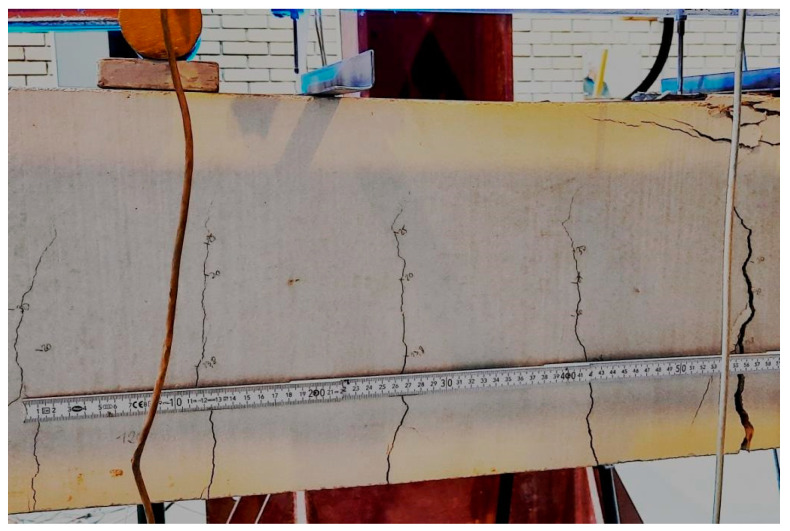
The cracking pattern captured by marking in RC beam.

**Figure 12 materials-15-02652-f012:**
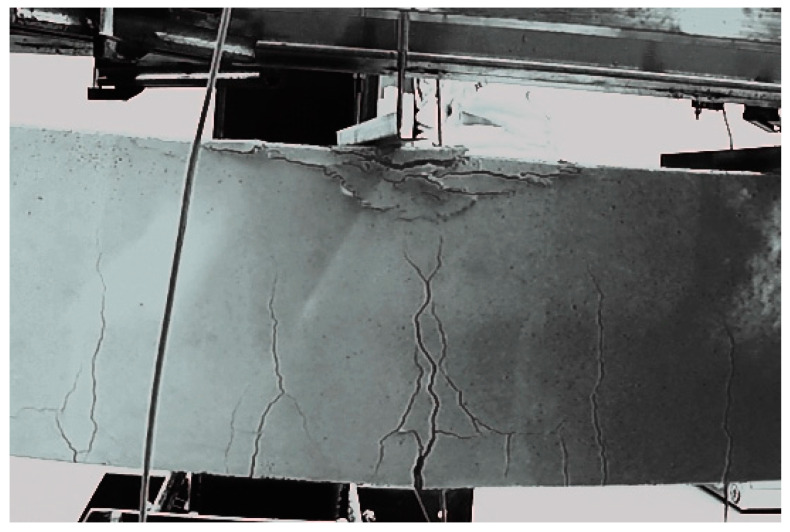
The cracking pattern at the midspan of the RC beam without polyurea coating.

**Figure 13 materials-15-02652-f013:**
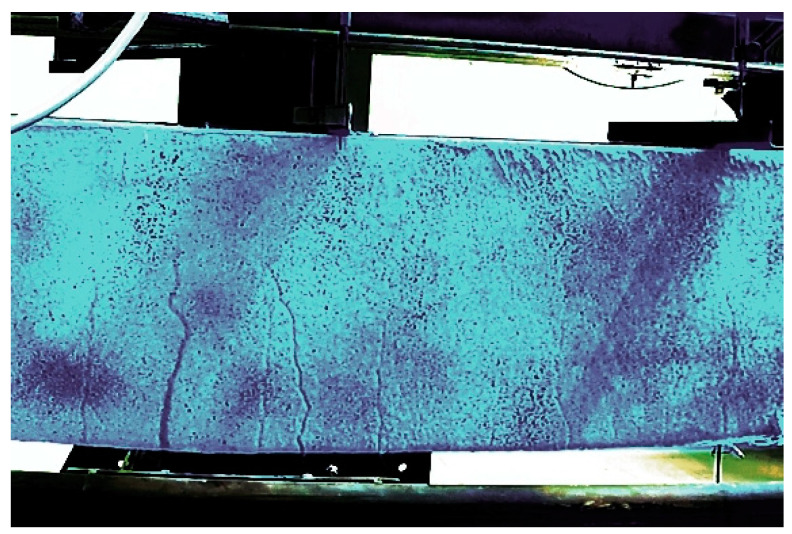
The cracking pattern at the midspan of the polyurea-coated RC beam.

**Figure 14 materials-15-02652-f014:**
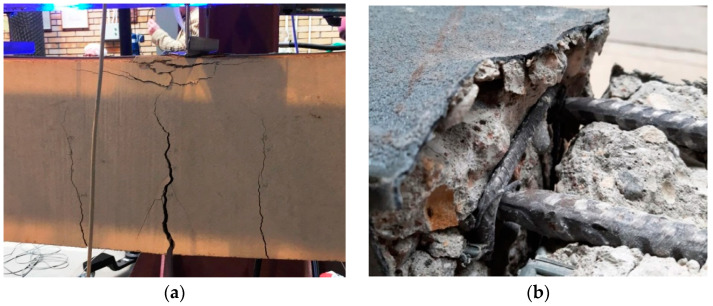
Failure mechanisms of the RC beams: (**a**) A beam without the polyurea coating; (**b**) A polyurea-coated beam.

**Figure 15 materials-15-02652-f015:**
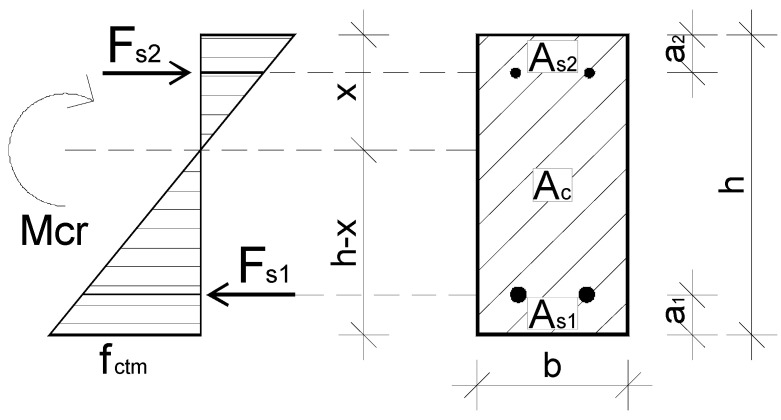
Theoretical concrete stress diagram—transition between phases I and II.

**Figure 16 materials-15-02652-f016:**
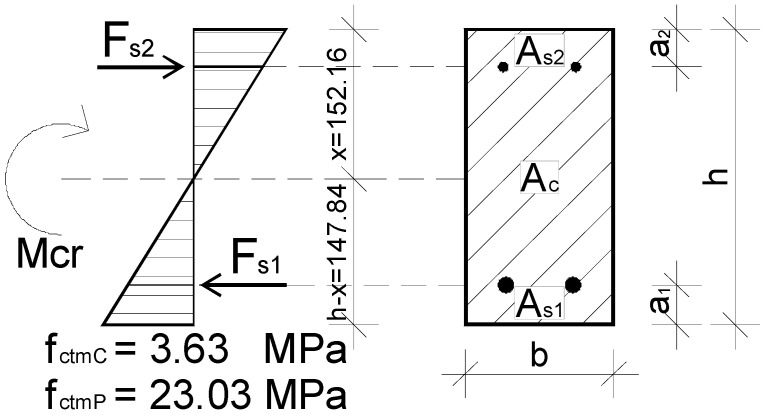
Actual concrete stress diagram—transition between phases I and II.

**Table 1 materials-15-02652-t001:** Strength properties of a polyurea coating.

Test Speed	Number of Tests	Tensile Strength	Engineering Strain	Young’s Modulus
(mm/min)	(-)	(MPa)	(%)	(MPa)
50	5	24.08	417	39.95
100	5	23.03	391	44.76

**Table 2 materials-15-02652-t002:** The summary of the main components of the concrete mix.

Component	Amount per 1 m^3^
(-)	(kg)
Sand (0–2 mm)	610
Gravel (2–8 mm)	470
Gravel (8–16 mm)	690
Cement CEM I 42.5	400
Water	100
Admixtures	3.3

**Table 3 materials-15-02652-t003:** Basic strength properties of the concrete.

Specimen Number	Specimen Dimensions	Compression Strength		Tensile Splitting Strength	
		Result	Average	Result	Average
(-)	(-)	(MPa)	(MPa)	(MPa)	(MPa)
01	150 × 150 × 150	71.01	72.35	-	-
02	150 × 150 × 150	72.56		-	
03	150 × 150 × 150	73.49		-	
04	150 × 150 × 150	-	-	3.54	3.63
05	150 × 150 × 150	-		3.83	
06	150 × 150 × 150	-		3.50	

**Table 4 materials-15-02652-t004:** Basic strength properties of reinforcing steel.

Rebar Diameter	Number of Tests	Lower Yield Stress	Tensile Strength	Young’s Modulus
(-)	(-)	(MPa)	(MPa)	(GPa)
#6	6	520.80	584.07	199.90
#10	6	535.10	647.60	200.57
#14	6	508.68	611.10	204.52

**Table 5 materials-15-02652-t005:** Description of test specimens.

Batch	Beam Designation	Average Coating Thickness	Load
(-)	(-)	(mm)	(-)
(1) beams used as control specimens	B.2.1	-	bending test
	B.2.2	-	bending test
	B.2.3	-	bending test
(2) polyurea-coated beams	P.2.1	2.5–3.0 *	bending test
	P.2.2	2.5–3.0 *	bending test
	P.2.3	2.5–3.0 *	bending test

* In our own course of research on polyurea coatings, it was observed that the thickness of the coating should be in the range of 2.5–3.0 mm to ensure adequate crack bridging efficiency in reinforced concrete members.

**Table 6 materials-15-02652-t006:** Summary of breaking forces for each batch of RC beams.

Batch	Beam Designation	Breaking Force Exerted by the Piston *	Average Breaking Force	Breaking Force Gain	Beam Deflection for F_max_
(-)	(-)	(kN)	(kN)	(kN/%)	(mm)
beams used as control specimens	B.2.1	92.2 *	97.9	-/-	43.0
	B.2.2	101.2 *			50.4
	B.2.3	100.3 *			48.6
polyurea-coated beams	P.2.1	103.1 *	102.5	+4.6 kN(+4.7%)	50.7
	P.2.2	103.2 *			44.7
	P.2.3	101.2 *			42.0

* Breaking force exerted by the piston is the maximum recorded value of the force in the main hydraulic piston during the examination of the reinforced concrete beam.

**Table 7 materials-15-02652-t007:** The results of calculations required to determine the height of the compression area.

Symbol	*E_s_*	*E_cm_*	*α_e_*	*A_c_*	*α_e_* × *A_s1_*	*α_e_* × *A_s2_*	*A_cs_*	*S_cs_*	*x*
Unit	(GPa)	(GPa)	(-)	(mm^2^)	(mm^2^)	(mm^2^)	(mm^2^)	(mm^3^)	(mm)
Value	200	32	6.25	45,000	1924.25	981.75	47,906	7,289,497	152.16
Comment	Standard value	Standard value	-	Cross-sectional area	Lower rebars 2#14	Upper rebars 2#10	Combined cross-section	Static moment	Height of compression area

**Table 8 materials-15-02652-t008:** The results of calculations performed to determine cracking moments.

Symbol	*h-x*	*I_cs_*	*W_c_*	*W_cs_*	*f_ctm_*	*M_cr1_*	*M_cr2_*
Unit	(mm)	(mm^4^)	(mm^3^)	(mm^3^)	(MPa)	(kNm)	(kNm)
Uncoated beam	147.84	374,172,588	2,250,000	2,530,972	3.63	8.17	9.19
Coated beam	147.84	374,172,588	2,250,000	2,530,972	23.03	51.82	58.29
Comment	Height of the tension area	Moment of inertia of the combined section	Section modulus	Section modulus of the combined section	Tensile strength	Cracking moment according to Equation (2)	Cracking moment according to Equation (3)

## Data Availability

Not applicable.

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
