# Peer review of "Influence of Polyurea Coatings on Low-Longitudinal-Reinforcement-Ratio Reinforced Concrete Beams Subjected to Bending"

_materials, 2022, doi:10.3390/ma15072652_

Round 1

Reviewer 1 Report

It is a interesting work.
1. The authors should give some study about which coating thickness is the best.
2. The quality of the figures and tables should be improved.

1)  Significance:
the results are interpreted appropriately
and all conclusions are justified and supported by the results.

2) Quality of Presentation:
the article is not written in an appropriate way.
the data and analyses not presented appropriately.
these need to be modified.

3)  Scientific Soundness:
3.1 the data are robust enough to draw the conclusions.
3.2 the methods, tools, software, and reagents are described with sufficient details to allow another researcher to reproduce the results.

Author Response

We would like to express our great appreciation to Editor and the Reviewers for the comments on our paper entitled ”Bending polyurea-coated reinforced concrete beams with a low longitudinal reinforcement ratio” which were very helpful for revising and improving them. We have studied these comments carefully and have made corresponding corrections that we hope will meet with your approval.

The changes in the revised manuscript are highlighted in yellow.

The responses to the Reviewers’ comments are provided below.

Answers for Reviewer 1 comments

We would like to thank you for your time spent to read and comment our paper. We hope that the answers below and corrections explain ambiguities and improve the clarity of the manuscript.

  • The authors should give some study about which coating thickness is the best.

Answer:
Thank you for your comment, it was explained below the Table 5 in the “Materials and Methods” section.

  • The quality of the figures and tables should be improved.

Answer:
The quality of the figures and tables was improved where it was possible.

  • The results are interpreted appropriately and all conclusions are justified and supported by the results.

Answer:
Thank you for your positive comment.

  • The article is not written in an appropriate way.

Answer:
The layout of the article was improved where it was possible.

  • The data and analyses not presented appropriately. These need to be modified.

Answer:

The data and analyses were improved where it was possible.

  • The data are robust enough to draw the conclusions.

Answer:
Thank you for your positive comment.

  • The methods, tools, software, and reagents are described with sufficient details to allow another researcher to reproduce the results.

Answer:
Thank you for your positive comment.

Reviewer 2 Report

This paper discusses the influence of polyurea coated reinforced concrete beams on their bending performance. Through the comparative experiment of coated polyurea and uncoated polyurea, the influence of polyurea on the bending performance of reinforced concrete beams is analyzed. The experiment in this paper is sufficient, and the conclusions obtained have a certain guiding significance for the research on the flexural performance of reinforced concrete beams. The article has the following aspects that need further discussion.

The paper is very interesting and well organized, but the author should clarify the following points:

  1. Line 240 shows the relationship between force and beam mid-span deflection, but can you explain the changing trend of each curve in the 0-10mmm deflection interval。
  2. Line 309 describes the test results in Figure 10. A large number of vertical cracks were observed, but an explanation should be given as to what caused this result.
  3. The author needs to increase the research on coating protective materials and point out the unique advantages of polyurea coating. The innovation of this paper should be highlighted from the experimental or theoretical point of view. In addition, the relevance and logic of the introduction part are not strong, so it is necessary to sort out the internal progressive relationship between relevant research contents and experimental conclusions, and enhance the level and logic of the paper.
  4. In this paper, the bending performance of polyurea coating on reinforced concrete beams with a low reinforcement ratio and the protective strengthening mechanism of polyurea coating on reinforced concrete beams are rarely described. It is suggested that the protection feasibility and unique advantages of polyurea coating materials under loading conditions should be elaborated from the perspective of theoretical research.
  5. The main innovation of this study is to explore the flexural performance of polyurea coating on reinforced concrete beams. The innovation of the study needs to be more clearly explained, for example, this special composite structure has better mechanical properties and process performance than other materials.
  6. There are few experiments in the article, which could not clearly explain the advantages of polyurea coating, and whether the thickness of polyurea coating could be further supplemented, so as to highlight the advantages of polyurea coating.
  7. The crack in Fig. 10 has a different location than the crack in Fig. 11. Is it the effect of polyurea? Please explain it.
  8. Line 379 “This is often a random value since the concrete structure is not always homogenous and RC elements can be of low quality in terms of workmanship.” Can authors give an approximate interval?
  9. Line 27 “The increase in load resulted in larger deflections but no side strains and no twist of the beams were observed during the tests”. Can authors give more detail on evidence?
  10. Provide more information about the loading mechanism of hydraulic pistons in thebeam bending experiment.
  11. How to measure and determine the breaking force of reinforced concrete beam in beam bending test.
  12. How to verify the theoretical calculation values of the cracking moment (Mcr1, Mcr2) of reinforced concrete beams coated with polyurea and uncoated with polyurea, and whether the experimental results can be given for comparison.
  13. There are too many overlapping parts of the lines in Figure 9 (b) and (c). It is recommended to give a partially enlarged view.
  14. Figures 10 and 11 compare the cracks between the two, but the author only shows that the polyurea layer covers the cracks with a width of 5mm, but does not give the maximum crack width of ordinary concrete beams, and it can be seen from the pictures. The crack patterns of the two beams seem to be different, whether it is because of the influence of the polyurea coating, I hope the author adds.
  15. It is recommended to mark the part name of the test bench in Figure 6, and also need to describe the model of the test stand.
  16. The coating thickness can be marked in Figure 4;
  17. The cracks in Figure 10 are not obvious in the picture, and the cracks can be outlined with software to make them more obvious.

Author Response

We would like to express our great appreciation to Editor and the Reviewers for the comments on our paper entitled ”Bending polyurea-coated reinforced concrete beams with a low longitudinal reinforcement ratio” which were very helpful for revising and improving them. We have studied these comments carefully and have made corresponding corrections that we hope will meet with your approval.

The changes in the revised manuscript are highlighted in yellow.

The responses to the Reviewers’ comments are provided below.

Answers for Reviewer 2 comments

We would like to thank you for your time spent to read and comment our paper. We hope that the answers below and corrections explain ambiguities and improve the clarity of the manuscript.

  • Line 240 shows the relationship between force and beam mid-span deflection, but can you explain the changing trend of each curve in the 0-10 kN deflection interval.

Answer:
Thank you for your comment, the describe of the changing trend of deflection curve in the 0-10 kN was added before the Figure 8 in the text.

  • Line 309 describes the test results in Figure 10 (now 12). A large number of vertical cracks were observed, but an explanation should be given as to what caused this result.

Answer:
A large number of vertical cracks were observed in this area of elements, because this is the zone of pure bending between two concentrated forces - according to the assumed static scheme of the experiment. This explanation was added in the text.

  • The author needs to increase the research on coating protective materials and point out the unique advantages of polyurea coating. The innovation of this paper should be highlighted from the experimental or theoretical point of view. In addition, the relevance and logic of the introduction part are not strong, so it is necessary to sort out the internal progressive relationship between relevant research contents and experimental conclusions, and enhance the level and logic of the paper.

Answer:
Thank you for your comment and suggestion. The presented work is a part of a complex research project entitled "Polyurea coatings as a possible structural reinforcement system", which also covers the subject of the unique properties of polyurea coatings and this aspect is analyzed in another research and publications (please check another authors’ works). The internal progressive relationship between relevant research contents and experimental conclusions was analyzed and sort out one more time, where it was possible.

  • In this paper, the bending performance of polyurea coating on reinforced concrete beams with a low reinforcement ratio and the protective strengthening mechanism of polyurea coating on reinforced concrete beams are rarely described. It is suggested that the protection feasibility and unique advantages of polyurea coating materials under loading conditions should be elaborated from the perspective of theoretical research.

Answer:
Thank you for your comment and suggestion. According to the abstract and the introduction to the article, the main subject of the study was the analysis of bent reinforced concrete elements covered with a polyurea coating. Failure mechanisms and cracking patterns of these specimens are described, and their bending strengths are compared. On this basis, the effect of the coating on bending strength, failure mechanism, cracking patterns and performance of the reinforced beams at the serviceability limit state (SLS) was examined and analyzed. The protection feasibility and unique advantages of polyurea coating materials under loading conditions are not the main subject of the work.

As mentioned earlier, the research presented in this article is a part of a complex research project entitled "Polyurea coatings as a possible structural reinforcement system", which also covers the subject of the unique properties of polyurea coatings and this aspect is analyzed in another research and publications.

  • The main innovation of this study is to explore the flexural performance of polyurea coating on reinforced concrete beams. The innovation of the study needs to be more clearly explained, for example, this special composite structure has better mechanical properties and process performance than other materials.

Answer:
Thank you for your comment. In contrast to traditional isolating materials, polyurea membranes have excellent functional properties, chemical resistance, and mechanical strength. The material properties of polyurea that make it usable in a wide range of applications and are at the same time its advantages include: fast reactivity and bonding (time saving), adherence to most building materials, high mechanical strength, effective crack bridging - experimental studies have shown that a coating can bridge cracks up to 5 mm in width. It was explained in the Conclusions section and of course this properties was analyzed in this study and another authors’ works and experiments.

  • There are few experiments in the article, which could not clearly explain the advantages of polyurea coating, and whether the thickness of polyurea coating could be further supplemented, so as to highlight the advantages of polyurea coating.

Answer:
Thank you for your comment, it was explained below the Table 5 in the “Materials and Methods” section and in the “Conclusions” section.

  • The crack in Fig. 10 (now 12) has a different location than the crack in Fig. 11 (now 13). Is it the effect of polyurea ? Please explain it.

Answer:
In the case of the polyurea-coated RC beams (specimens P.2.1 to P.2.3), the largest cracks were observed at the beam midspan. The polyurea coating efficiently covered the surface cracking to the extent that only wide cracks could be seen. The crack bridging by the coating generated a difference in the visual perception of the crack pattern on elements covered with polyurea coating in relation to uncoated beams (Figure 10 and 11, now 12 and 13), because in this case smaller scratches were not noticeable to the unaided eye at all.

  • Line 379 “This is often a random value since the concrete structure is not always homogenous and RC elements can be of low quality in terms of workmanship.” Can authors give an approximate interval ?

Answer:
Approximate interval of this value is difficult to clearly indicate, because it is a multi-parameter phenomenon.

  • Line 27 “The increase in load resulted in larger deflections but no side strains and no twist of the beams were observed during the tests”. Can authors give more detail on evidence ?

Answer:
Description was supplemented with photos showing beams under loading as views along the elements as evidence of this statement – Figure 9.

  • Provide more information about the loading mechanism of hydraulic pistons in the beam bending experiment.

Answer:
The RC beams were loaded by a steel traverse which was oriented symmetrically to the beam axis and produced load in the form of two concentrated forces 100 cm apart. Force was applied to the steel traverse via a centrally located hydraulic piston. The hydraulic piston applied the appropriate pressure by means of a hydraulic pump, the pressure of which was increased in steps to obtain step increases in the value of the force on the piston. Reinforced concrete beams were loaded with step force increments of 5 kN or 10 kN. A force gauge was mounted between the hydraulic piston and the steel traverse in order to double verify the current value of the load on the reinforced concrete beam. Description with this information is given in the “Test stand” section.

  • How to measure and determine the breaking force of reinforced concrete beam in beam bending test.

Answer:
Breaking force exerted by the piston is the maximum recorded value of the force in the main hydraulic piston during the examination of the reinforced concrete beam. This information is added below Table 6.

  • How to verify the theoretical calculation values of the cracking moment (Mcr1, Mcr2) of reinforced concrete beams coated with polyurea and uncoated with polyurea, and whether the experimental results can be given for comparison.

Answer:
The cracking pattern was observed during the tests on a continuous basis using two cameras from both sides of the RC beams. For selected specimens, the cracking pattern on the surface of the elements was also captured by marking, near each crack, the load value at which it formed. The cracking pattern captured by marking was authors’ verification of the theoretical calculation values of the cracking moment (Mcr1, Mcr2). In the case of the polyurea-coated RC beams, the polyurea coating efficiently covered the surface cracking to the extent that only wide cracks could be seen. In this case, authors used ARAMIS system for verification of the cracking pattern, but authors’ results with ARAMIS system outcomes will be the topic of another article. The photo with manual scratch measurements also was added in the “Component cracking” section.

  • There are too many overlapping parts of the lines in Figure 9 (b) and (c) (now Figure 10b and 10c). It is recommended to give a partially enlarged view.

Answer:
Thank you for your recommendation, the partially enlarged views were added in Figure 9 (now 10).

  • Figures 10 and 11 (now 12 and 13) compare the cracks between the two, but the author only shows that the polyurea layer covers the cracks with a width of 5mm, but does not give the maximum crack width of ordinary concrete beams, and it can be seen from the pictures. The crack patterns of the two beams seem to be different, whether it is because of the influence of the polyurea coating, I hope the author adds.

Answer:
In the case of the polyurea-coated RC beams (specimens P.2.1 to P.2.3), the largest cracks were observed at the beam midspan. The polyurea coating efficiently covered the surface cracking to the extent that only wide cracks could be seen. The crack bridging by the coating generated a difference in the visual perception of the crack pattern on elements covered with polyurea coating in relation to uncoated beams (Figure 10 and 11, now 12 and 13), because in this case smaller scratches were not noticeable to the unaided eye at all. The photo with manual scratch measurements also was added in the “Component cracking” section.

  • It is recommended to mark the part name of the test bench in Figure 6, and also need to describe the model of the test stand.

Answer:
Thank you for your suggestion, the elements markings were added in Figure 6 and describe of the test stand was supplemented.

  • The coating thickness can be marked in Figure 4.

Answer:
Thank you for your comment, it was done.

  • The cracks in Figure 10 (now 12) are not obvious in the picture, and the cracks can be outlined with software to make them more obvious.

Answer:
The quality of Figure 10 (now 12) was improved.

Reviewer 3 Report

The manuscript deals with the experimental studies on the flexural behaviour of RCC beams improved with polyuria coating. The performance of RCC beams improved with coating is more pronounced in the proposed research. There is a lack of consistency between the proposed objectives and work done. Therefore I propose the manuscript needs major revision before the acceptance, so that the reader can get better benefits out of the results.  Title is little confusing, revise the title for better understanding  In the Abstract section avoid writing “project …………….”  It is unclear that how the coating will contribute to retain/enhance the bending capacity??  Also, the serviceability limit state in terms of durability governs the beam members under corrosion or fire. The performance of said coating to be tested after the exposure to mentioned conditions. Justify the reasons for this with strong background  It is quite difficult to accept the importance of this coating in terms of improving the bending performance, as it is the commercial product there is a lack of novelty exist  Why did the author use bent RCC beams??  The tensile strength of coating is 23MPa, it is almost 22 times lesser than the original yield strength of rebars, and how the coating will contribute during the failure by enhancing the elastic properties?  Why the authors use the term low longitudinal reinforcement ratio? Is there any significance, discuss  The increase in load carrying capacity and yield deformation of coated and uncoated specimens to be discussed in detail with appropriate reasons/evidence  The increase in ultimate load for a coated beam seems to be less than 5%, may not be acceptable for performance evaluation, discuss  Details of crack, pattern, width, and propagation is not explained in a better manner  There are explanations in terms of bending stiffness but I can’t find any table/graph indicating such values  In the conclusion section, the following statement is given but does not have any experimental results on this; authors should avoid writing generic statements in conclusion. Also some statements are explaining as if the experiments are conducted to assess the long term performance, but the study is limited to short term loading Check the following statements The polyurea coating efficiently covers cracks and protects the element against penetration by corrosive fluids (water, air, and chemical compounds); When the polyurea coating is applied, it brings considerable benefits when RC beams are used at the serviceability limit state because the coating covers cracks and prolongs the service life of the elements  I strongly suggest the authors to recheck the manuscript, specific to the write-ups considering the serviceability and durability of beams. Better to rephrase those based on the exact application of the proposed coating  Rewrite the conclusion section; it is not specific based on the proposed results  Introduction section is written based on the corrosion performance of coated beams, but the proposed study is different. Check the introduction section and rewrite based on the proposed objectives  Research gap, discussion should be improved based on the novelty of the study  Clearly describe the limitations of the study, so that the reader can understand the objectives

Author Response

We would like to express our great appreciation to Editor and the Reviewers for the comments on our paper entitled ”Bending polyurea-coated reinforced concrete beams with a low longitudinal reinforcement ratio” which were very helpful for revising and improving them. We have studied these comments carefully and have made corresponding corrections that we hope will meet with your approval.

The changes in the revised manuscript are highlighted in yellow.

The responses to the Reviewers’ comments are provided below.

Answers for Reviewer 3 comments

We would like to begin by expressing sincere thanks for the reviewer's recognition of presented work. The work has been strengthened in accordance with the reviewer's suggestions.

  • Title is little confusing, revise the title for better understanding.

Answer:

The authors admit that the title can be a bit confusing and propose the following change to the title: “Influence of polyurea-coating on low longitudinal reinforcement ratio reinforced concrete beams subjected to bending”.

  • In the Abstract section avoid writing “project …………….”

Answer:

Thank you for your comment, it was included in the Abstract section.

  • It is unclear that how the coating will contribute to retain/enhance the bending capacity ?

Answer:
For clarification, the authors added in Introduction section an explanation:

“It was stated that the proposed method of covering a reinforced concrete beam with a polyurea layered coating does not significantly increase the bending capacity, but it enables the improvement of the performance characteristics of a reinforced concrete beam in its normal and exceptional condition, consisting in ensuring the possibility of cyclic loading and unloading of beam elements without visible loss of load capacity.

The use of a polyurea coating on the surfaces of reinforced concrete beam elements allows for safe relief of the structure after a load exceeding its limit load capacity. Such a situation may take place when the structure is operating in a failure state. Thus, the use of a polyurea coating can significantly improve the safety of the entire structure and its users. The new characteristics of the bending element, obtained in this way, enable the element to be used after exceeding the permissible deformations, while maintaining an acceptable level of safety ”.

  • Also, the serviceability limit state in terms of durability governs the beam members under corrosion or fire. The performance of said coating to be tested after the exposure to mentioned conditions. Justify the reasons for this with strong background.

Answer:
Unfortunately in the paper authors do not referred to the situation in which the beam member is under corrosion or fire. The authors agree with the Reviewer's comment, but they are convinced that the signalled problem belongs to a completely different class of scientific problems and may be the subject of other extended research. For this reason, it is neither the main issue not the background to the conducted research.

  • It is quite difficult to accept the importance of this coating in terms of improving the bending performance, as it is the commercial product there is a lack of novelty exist.

Answer:

The essence of the invention is to increase the crack-bridging efficiency and to enable the improvement of the performance of a reinforced concrete beam in normal and failure state. The application of polyurea in few layers made it possible to periodically load and relieve the beam elements without visible loss of their load capacity and with a satisfactory level of crack bridging. An epoxy resin primer is applied to the sanded surface of the beam. The preliminary stage is to eliminate substances that adversely affect the adhesion of the coating, e.g. oils, greases, cement laitance and any other loose concrete particles. The primed beam is sprinkled with dried quartz sand with a grain size of 0.3 - 0.8 mm for the best adhesion of the coating. Then the polyurea coating is applied in few layers by spraying. the first layer is applied directly to the concrete surface, the second layer is applied directly to the first layer perpendicular to the direction of application of the first layer and the third layer directly to the second layer parallel to the direction of application of the first layer, the average total thickness of the three layers of coating is 2.5-3.0 mm. This method of application will ensure a fully tight and continuous coating, which significantly increases the effectiveness of crack bridging in reinforced concrete elements.

  • Why did the author use bent RCC beams ?

Answer:

The authors of this work this time focused on the analysis of bent reinforced concrete (RC) beams reinforced with a polyurea coating as one of the most commonly met structural element. Previously, the concrete rings coated with polyurea were analysed and experimentally tested. The research presented in this paper is therefore only a part of a broad research program which is focused on investigation the effect of polyurea on the behavior of various reinforced concrete, steel or timber components.

  • The tensile strength of coating is 23 MPa, it is almost 22 times lesser than the original yield strength of rebars, and how the coating will contribute during the failure by enhancing the elastic properties ?

Answer:

Taking into account the experience gained during laboratory tests of reinforced concrete rings and the large influence of the polyurea coating on the elastic operation of these rings under load, it was decided to check whether the bending beams would behave similarly after loading and unloading. In the case of bent beams, the elastic behaviour turned out to be less visible than in the case of reinforced concrete rings, however, similarly to them, also after reaching a high level of load, it was possible to unload the element and load it again.

  • Why the authors use the term low longitudinal reinforcement ratio? Is there any significance, discuss.

Answer:

As mentioned earlier, the research presented in this article is part of a broader research program. This time, the analysis of beams with a low reinforcement ratio was limited due to the fact that the external load favors the formation of a specific crack pattern, i.e. a large number of cracks with small widths. High reinforcement beams may behave differently and the effect of using a polyurea coating may be relatively slightly different. This aspect will be analyzed in further research (laboratory tests are already done).

  • The increase in ultimate load for a coated beam seems to be less than 5%, may not be acceptable for performance evaluation, discuss.

Answer:

As it was already mentioned, the main purpose of using the polyurea coating was not to increase the beam's bending capacity in the strict sense, but to increase beam’s integrity and durability in the context of cyclic loading and failure state. In the case of a beam not protected with a polyurea coating, the exceeding the permissible crack opening is the basic failure mechanism. In the case of a beam with a polyurea coating, the crack opening does not result in the destruction of the element in the classical sense, because the coating ensures the integrity of the element even after exceeding the permissible crack opening.

  • Details of crack, pattern, width, and propagation is not explained in a better manner.

Answer:

Regarding to reviewer’s comment, the an appropriate explanation as well as additional drawing and describe were included in the text.

  • There are explanations in terms of bending stiffness but I can’t find any table/graph indicating such values.

Answer:

The behaviour of reference beams and polyurea reinforced beams under load and the bending stiffness of these beams are discussed in the description of Figure 8. This drawing shows, in our opinion, the behavior of these beams sufficiently under the bending moment.

Reviewer 4 Report

The manuscript materials-1640869 is presented on the “Bending polyurea-coated reinforced concrete beams with a low longitudinal reinforcement ratio”. In my opinion, the manuscript was generally well written and displayed interesting work and relevant to this Journal. I recommend accepting this manuscript for publication after minor revisions as follows:

  • The author mentioned that the “Polyurea is the reaction product of two components” what are the comments used for the preparation of Ployurea?
  • What is the optimum temperature for spraying polyura in this work and how the author control the temperature?
  • What is the economic vision for using Ployurea in construction?

Author Response

We would like to express our great appreciation to Editor and the Reviewers for the comments on our paper entitled ”Bending polyurea-coated reinforced concrete beams with a low longitudinal reinforcement ratio” which were very helpful for revising and improving them. We have studied these comments carefully and have made corresponding corrections that we hope will meet with your approval.

The changes in the revised manuscript are highlighted in yellow.

The responses to the Reviewers’ comments are provided below.

Answers for Reviewer 4 comments

We would like to begin by expressing sincere thanks for the reviewer's recognition of presented work. The work has been strengthened in accordance with the reviewer's suggestions.

  • The author mentioned that the “Polyurea is the reaction product of two components” what are the comments used for the preparation of Ployurea ?

Answer:
The tests were determined with used aromatic polyurea as the most common type of coating utilized in the construction industry. Aromatic polyureas are derived from methylenediphenyl diisocyanate (MDI) which form a stiff chain section. The precision of mixing and dosing the components plays a prominent role in the spray showering process of polyurea. The parameters of spraying (temperature and pressure in the device) have to be rigorously controlled. They have to comply with the norms given by the manufacturer in the Product Data Sheet (DPS). Moreover the appropriate voluminal and weight ratios of the components have to be continuously controlled before and during the process of spraying. The text was supplemented about this information in the “Polyurea” section.

  • What is the optimum temperature for spraying polyura in this work and how the author control the temperature ?

Answer:
The optimum temperature for spraying polyurea in this work is:

  • + 750C for ingredients and it was checked by special set of machine called aggregate for spraying polyurea,
  • + 150C for surface of concrete and atmospheric and it was checked by handheld thermometer.

The text was supplemented about this information in the “Beams with polyurea outer layer” section.

  • What is the economic vision for using Polyurea in construction ?

Answer:
The final cost of applying the polyurea coating largely depends on humidity, quality, and level of damage of the existing concrete surface. The cost of such a service is proportional to the scope of preparation processes required to achieve a proper condition of the surface so that appropriate epoxy resin-based primer can be applied. Estimated costs of polyurea coating application including surface preparation are between:

  • From 75 EUR/m2 net for dry concrete surface without defects and cracks;
  • To 160 EUR/m2 net for wet concrete with deep defects and cracks.

Reinforced concrete elements are a specific type of structural components very often designed to specific conditions and inside buildings. This is why any repair of such components requires making a lot of works and involves severe difficulties inside buildings. The use of polyurea coatings to repair existing reinforced concrete elements and/or to improve their properties can prove a very cost-effective renovation method. This solution requires no many works and the components can be repaired inside buildings, which can be highly cost-effective both in financial and organizational terms. The text was supplemented about this information in the “Cost analysis of polyurea coating application on reinforced concrete elements” new section.

Round 2

Reviewer 2 Report

The revisions are satisfying and I am convinced about the publication of this submission.

Reviewer 3 Report

There are some improvements in the content